🔓 | **Open Peer Review** | Epidemiology | Research Article

# A novel simian adenovirus-vectored COVID-19 vaccine elicits effective mucosal and systemic immunity in mice by intranasal and intramuscular vaccination regimens

Panli Zhang,[1,2] Shengxue Luo,[3] Peng Zou,[1,2] Qitao Deng,[1,2] Cong Wang,[1,2] Jinfeng Li,[4] Peiqiao Cai,[5] Ling Zhang,[1] Chengyao Li,[1] Tingting Li[1]

**ABSTRACT**  The failure of COVID-19 vaccines to prevent SARS-CoV-2 infection and transmission, a possibly critical reason was the lack of protective mucosal immunity in the respiratory tract. Here, we evaluated the effects of mucosal and systemic immunity from a novel simian adenovirus-vectored COVID-19 vaccine (Sad23L-nCoV-S) in mice in comparison with Ad5-nCoV-S by intranasal (IN) drip and intramuscular (IM) injection vaccinations. As good as the well-known Ad5-nCoV-S vaccine, a single-dose IN inoculation of $1 \times 10^9$ PFU Sad23L-nCoV-S vaccine induced a similar level of IgG S-binding antibody (S-BAb) and neutralizing antibody (NAb) and higher IgA in serum, while IN route raised significantly higher IgG and IgA S-BAb and NAb in bronchoalveolar lavage (BAL), and specific IFN-γ secreting T-cell response in lung compared with IM route, but lower T-cell response in spleen. By prime-boost vaccination regimens with different combinations of IN and IM inoculations of Sad23L-nCoV-S vaccine, the IN-involved vaccination stimulated higher protective mucosal or local immunity in BAL and lung, while the IM-involved immunization induced higher systemic immunity in serum and spleen. A long-term sustained mucosal and systemic NAb and T- cell immunity to SARS-CoV-2 was maintained at high level over 32 weeks by prime-boost vaccination regimens with IN and IM routes. In conclusion, priming or boosting immunization with IN inoculation of Sad23L-nCoV-S vaccine could induce effective mucosal immunity and in combination of IM route could additionally achieve systemic immunity, which provided an important reference for vaccination regimens against respiratory virus infection.

**IMPORTANCE**  The essential goal of vaccination is to generate potent and long-term protection against diseases. Several factors including vaccine vector, delivery route, and boosting regimen influence the outcome of prime-boost immunization approaches. The immunization regimens by constructing a novel simian adenovirus-vectored COVID-19 vaccine and employing combination of intranasal and intramuscular inoculations could elicit mucosal neutralizing antibodies against five mutant strains in the respiratory tract and strong systemic immunity. Immune protection could last for more than 32 weeks. Vectored vaccine construction and immunization regimens have positively impacted respiratory disease prevention.

**KEYWORDS**  adenoviral vector, intranasal immunization, mucosal immunity, respiratory virus, vaccination regimen

During the past 3three years, the severe acute respiratory syndrome coronavirus 2 (SARS-CoV-2) infection caused a global pandemic of coronavirus disease-19 (COVID-19) and led to heavy losses of human life and the world economy (1, 2). Multiple vaccines including mRNA, adenoviral vector, inactivated virus, and recombinant

Address correspondence to Chengyao Li, chengyaoli@hotmail.com, or Tingting Li, appleting-007@163.com.

Panli Zhang and Shengxue Luo contributed equally to this article. Author order was determined by drawing straws.

The authors declare no conflict of interest.

See the funding table on p. 14.

subunit vaccines have been developed and approved for emerging use (3). Current COVID-19 vaccines were administered to individuals mostly by the intramuscular (IM) route, especially for mRNA vaccines, and were likely to induce a better systemic immune response, which insufficiently protected virus infection in the upper respiratory tract (4–6), and breakthrough infections were commonly seen (7, 8).

Nasal vaccines hold a superior position against infections such as SARS-CoV-2, whose invasion occurs majorly via the nasal mucosa (9). With the large-scale prevalence of Omicron variant with greater transmissibility, there is a greater need than ever for mucosal vaccines. Such protective immunity includes secretory IgA and resident memory T cells (10, 11). As compared to IgG, the secreted IgA molecules may contribute to superior virus neutralization potency, and possibly broad neutralization of antigenically diverse viruses (12).

Adenoviruses as vectors are relatively easy to penetrate the respiratory mucosal barrier, achieve antigen delivery, and provoke respiratory mucosal immune responses. In addition, its ability to generate such responses at the mucosal sites of pathogen entry makes it a preferred choice for intranasal (IN) vaccination (13). Nasal inoculation of adenovirus‐vectored vaccines, either as prime or boost vaccination in combination with IM mRNA vaccine, induced a high level of local antibodies that could prevent viral replication in the upper respiratory tract of mice or monkeys (14, 15). Specifically, in murine and ferret models, a human adenovirus-vectored COVID-19 vaccine (Ad5-nCOV) with a single mucosal inoculation provided protection for the upper and lower respiratory tracts against SARS-CoV-2 infection (16). Moreover, aerosolized Ad5-nCoV as a booster significantly enhanced systemic immunity and stimulated mucosal immunity in a phase I clinical trial, which was approved for the use in China (17, 18). Intranasal administration of ChAdOx1 nCoV-19 protected against the SARS-CoV-2 challenge in hamsters and non-human primates (NHPs) (19). However, the clinical phase I trial showed a low proportion of individuals producing mucosal antibodies after a single dose of IN vaccination in a small sample size. While after two doses of IN inoculations, only a minority of participants (4/13) had a significant response (20). These suggested the need for a combination of different immunization routes. Previous studies showed that IN route could compensate for the limitation of IM route that provoked weak mucosal immunity (21), but the induced cellular immune response was relatively poor (22).

We have previously developed a COVID-19 vaccine (Sad23L-nCoV-S) based on a simian adenovirus type 23 vector (Sad23L) carrying the full-length S gene of SARS-CoV-2 (23). Our previous studies have shown that IM inoculation with this vaccine elicited strong humoral and cellular immune responses in C57BL/6 and BALB/C mice, cats, and rhesus monkeys (23, 24). The combination of IM and IN inoculations, which induce the durable systemic IgG, local mucosal antibodies and memory T cells, ultimately leads to a long-lasting protective immunity, achieving that the whole is greater than the sum of its parts, most likely one of the ideal vaccination strategies for obtaining prolonged immunity (25). In this study, we evaluated the systemic and mucosal immune effects of novel Sad23L-nCoV-S vaccine from different combinations of IN and IM inoculations in comparison with Ad5-nCoV-S vaccine in mice.

## RESULTS

### Production and identification of Sad23L-nCoV-S and Ad5-nCoV-S vaccines

Sad23L-nCoV-S vaccine carrying the full-length SARS-CoV-2 spike gene was constructed previously (23). Ad5-nCoV-S vaccine was used as a control, of which the same form vaccine was approved for emergency use as an aerosol booster in China (17). Both Sad23L-nCoV-S and Ad5-nCoV-S vaccines were produced and purified from the HEK293 cell cultures, and then were negatively stained with 2% phosphotungstic acid. The vaccine strains were observed by transmission electron microscope (TEM), which presented an entire and classic morphology of adenovirus (Fig. 1a and b). The high expression of SARS-CoV-2 S protein was detected in both adenovirus-vectored vaccines infected HEK293 cells by the IFS and Western blot, respectively (Fig. 1c and d). These

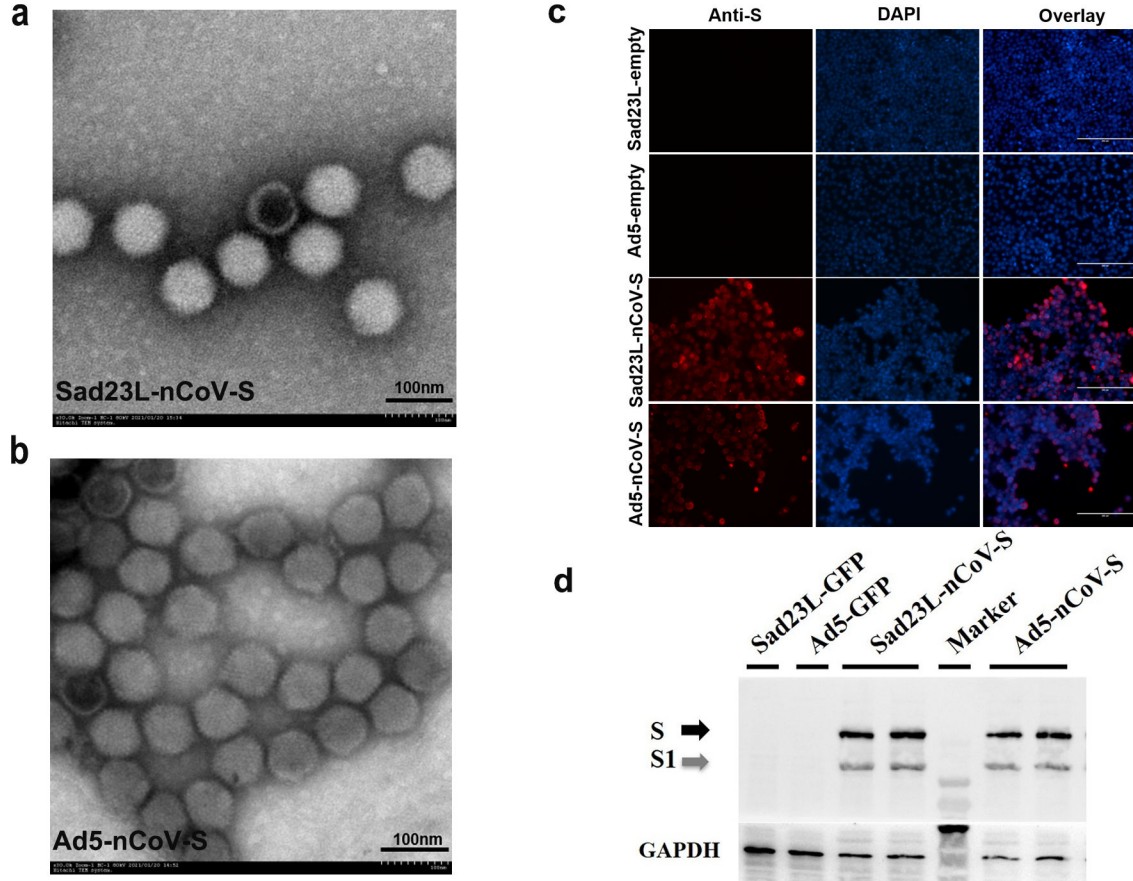

**FIG 1** Production of Sad23L-nCoV-S and Ad5-nCoV-S vaccines. (a) Images of Sad23L-nCoV-S and (b) Ad5-nCoV-S vaccine stains by negatively stained with 2% phosphotungstic acid under transmission electron microscope (TEM). (c) Immunofluorescence staining for SARS-CoV-2 S protein in adenovirus-vectored vaccine infected HEK293A cells. (d) Western blot analysis for Sad23L-nCoV-S or Ad5-nCoV-S vaccine infected HEK-293A cells by rabbit polyclonal antibody specific to RBD of SARS-CoV-2.

results showed that Sad23L-nCoV-S and Ad5-nCoV-S vaccines could effectively produce SARS-CoV-2 S protein in the vaccine-infected cells.

## Evaluation of immune effects by a single-dose intranasal inoculation of Sad23L-nCoV-S vaccine in mice

The immunogenicity of a single dose of $10^9$ PFU Sad23L-nCoV-S vaccine was examined by inoculation via IN drip or IM injection in comparison with Ad5-nCoV-S vaccine in mice (Fig. 2a). After 4 weeks post-immunization, the serum and BAL samples were collected to measure IgG and IgA S1-BAb or S2-BAb titers by enzyme-linked immunosorbent assay (ELISA) and NAb titers ($IC_{50}$) by pVNT, respectively (Fig. S1). From IN inoculated mice with Sad23L-nCoV-S vaccine, serum IgG S1-BAb and S2-BAb titers ($10^{4.37}$ and $10^{3.54}$) were detected in high level, but slightly lower than those from IM injected mice ($10^{4.63}$ S1-BAb and $10^{4.00}$ S2-BAb) (Fig. 2b); serum IgA S1-BAb and S2-BAb titers ($\geq 10^{3.23}$) were significantly higher than those ($\leq 10^{2.03}$) from IM injected mice ($P < 0.001$; Fig. 2c); BAL IgG ($10^{2.57}$ and $10^{2.33}$) and IgA S1-BAb and S2-BAb activities ($10^{2.30}$ and $10^{2.06}$) were detected, but not from IM injected mice (Fig. 2d and e). From IN immunized mice, serum NAb titer was measured as $IC_{50}$ $10^{2.97}$ against wild-type strain or $IC_{50}$ $10^{1.83}$ against Omicron strain of pseudovirus, similar to serum NAb titers from IM immunized mice and convalescent serum of COVID-19 patients ($P > 0.05$; Fig. 2f), while BAL NAb was only detected at low level ($IC_{50}$ $10^{1.34}$) against wild-type strain of pseudovirus from IN immunized mice (Fig. 2g). A single dose of Ad5-nCoV-S vaccine was compared in parallel with Sad23L-nCoV-S

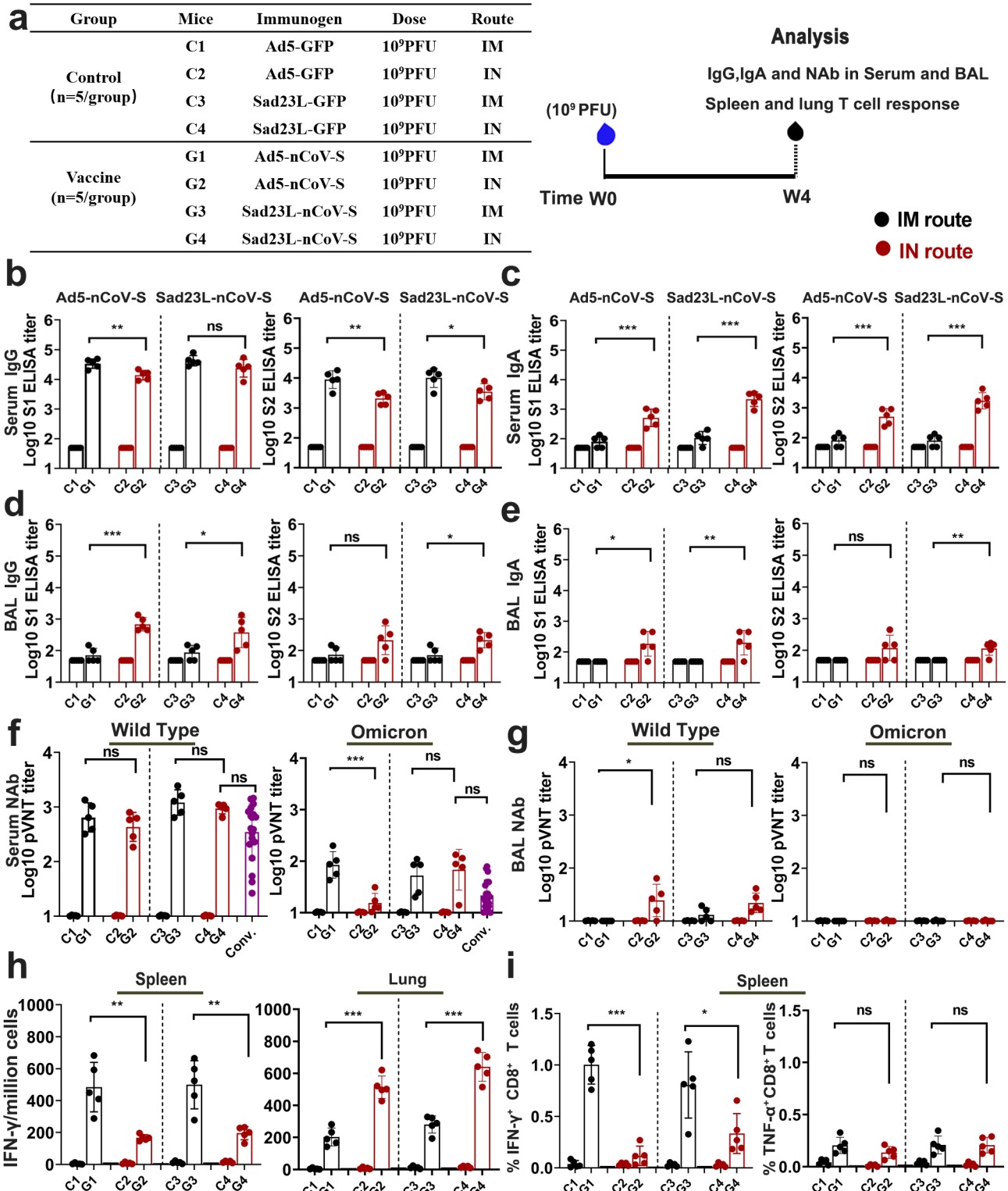

**FIG 2** Immune effectiveness of a single dose of Sad23L-nCoV-S or Ad5-nCoV-S vaccine in mice via IM or IN inoculation. (a) Group for control or vaccine, mice groups (C1–C4; G1–G4; $n = 5$), immunogen for immunization, dose and inoculation route and detection. (b–g) Antibody reactivity in serum and bronchoalveolar lavage (BAL) from intramuscular (IM) or intranasal (IN) inoculated mice by enzyme-linked immunosorbent assay (ELISA) or pseudovirus neutralization test (pVNT). (h) IFN-γ secreting T-cell response in spleen and lung cells was detected by enzyme-linked immunospot (ELISpot). The reacted T cells were presented as spot-forming cells (SFCs)/million cells. (i) IFN-γ or TNF-α expressing $CD8^+$ T-cell response in spleen was measured by flow cytometry. Data were shown as mean ± SEM (standard errors of means). $P$ values were analyzed by unpaired two-tailed $t$ test and Mann-Whitney test. Statistically significant differences were indicated with asterisks (*$P < 0.05$, **$P < 0.01$, and ***$P < 0.001$). ns, not significant ($P > 0.05$).

vaccine for IN immunization of mice, which presented a same pattern for S-BAb and NAb responses between both vaccines (Fig. 2b through g).

To assess the antigen-specific T-cell responses in mice induced by IN or IM inoculation of Sad23L-nCoV-S in comparison with Ad5-nCoV-S vaccine, splenic and lung cells were harvested to analyze T-cell response by the enzyme-linked immunospot (ELISpot) or flow cytometry (Fig. 2a). IN immunization of Sad23L-nCoV-S vaccine induced lower specific IFN-γ-secreting T-cell response in spleen but higher in lung (196 versus 640 SFCs/$10^6$ cells), while IM immunization stimulated higher specific IFN-γ-secreting T-cell response reversely in spleen but lower in lung (499 versus 281 SFCs/$10^6$ cells), respectively (Fig. 2h). Moreover, IN inoculation of Sad23L-nCoV-S vaccine raised relatively lower frequency of IFN-γ$^+$ CD8$^+$ T-cell response (0.33%) than IM immunization in spleen (0.81%, $P <$ 0.05), while both IN and IM immunizations induced TNF-α$^+$ CD8$^+$ T-cell responses varied insignificantly at low level in spleen (Fig. 2i). For comparison, IN or IM inoculation of Ad5-nCoV-S vaccine showed a dynamic pattern of T-cell responses consistent with Sad23L-nCoV-S in spleen and lung of mice (Fig. 2h and i).

Overall, as good as Ad5-nCoV-S vaccine, a single-dose IN or IM immunization of Sad23L-nCoV-S vaccine induced similar level of serum IgG S-BAb or NAb reactivity, while IN inoculation had stronger serum IgA, BAL S-BAb, or NAb reactivity, and higher specific T-cell response in lung but lower in spleen of immunized mice compared to IM route.

## Antibody reactivity in mice induced by prime-boost vaccination regimens with different combinations of intranasal and intramuscular inoculations of Sad23L-nCoV-S vaccine

In order to establish a proper prime-boost immunization regimens for optimizing immune efficacy with circulating and local mucosal immunity, four combinations of IM/IM, IN/IN, IM/IN, and IN/IM inoculations of Sad23L-nCoV-S vaccine were examined (Fig. 3a). All four vaccination regimens elicited high levels of serum IgG S1-BAb and S2-BAb (>$10^{4.51}$; Fig. 3b), IgA excepting IM/IM (G5) (>$10^{3.34}$; Fig. 3c), and NAb (IC$_{50}$ >$10^{3.45}$) higher than those in convalescent serum from COVID-19 patients (Fig. 3d), of which IN/IM and IM/IN regimens presented the better neutralizing activity (IC$_{50}$ $10^{4.08}$ and $10^{3.83}$, respectively). In BAL samples from prime-boost vaccinated mice, the IN/IN, IM/IN, and IN/IM immunization regimens containing IN inoculation of vaccine showed satisfactory IgG and IgA S-BAb (S1-BAb and S2-BAb) and NAb responses, of which IN/IN appeared relatively stronger in mice, while IM/IM regimens did not raise local mucosal antibody response (Fig. 3e through g). Strong correlation between IgG, IgA, and NAb reactivity in BAL was observed ($P <$ 0.0001, $R2 =$ 0.844–0.878; Fig. S2), suggesting the level of mucosal IgA associated with neutralization potency against this respiratory SARS-CoV-2 infection.

Five major variants of pseudotyped SARS-CoV-2 were generated for examining neutralizing efficacy of NAb in serum and BAL from above prime-boost vaccinated mice. Serum NAb activity from four groups of vaccinated mice was generally high against Alpha (>IC$_{50}$ $10^{3.13}$), Beta (>IC$_{50}$ $10^{2.54}$), Gamma (>IC$_{50}$ $10^{3.16}$), Delta (>IC$_{50}$ $10^{3.02}$), and Omicron (>IC$_{50}$ $10^{1.81}$) variants, of which all were significantly higher than those from convalescent serum samples of COVID-19 patients to five pseudoviruses ($P <$ 0.001; Fig. 4a). Of them, serum NAb titers appeared to be the highest in mice with IN/IM vaccination regimens (Fig. 4a). BAL NAb activity was detected at relatively low level from four regimens immunized mice (IC$_{50}$ $10^{1.05}$–$10^{2.13}$), excepting for no response from IM/IM inoculated mice (Fig. 4b). Compared to a single immunization (IN or IM), the combination of IM and IN inoculations increased neutralizing antibodies (Fig. 4). These data showed that prime-boost immunization regimens, in particular, with IN inoculation, could produce effectively mucosal as well as systemic neutralization potency to the major variants.

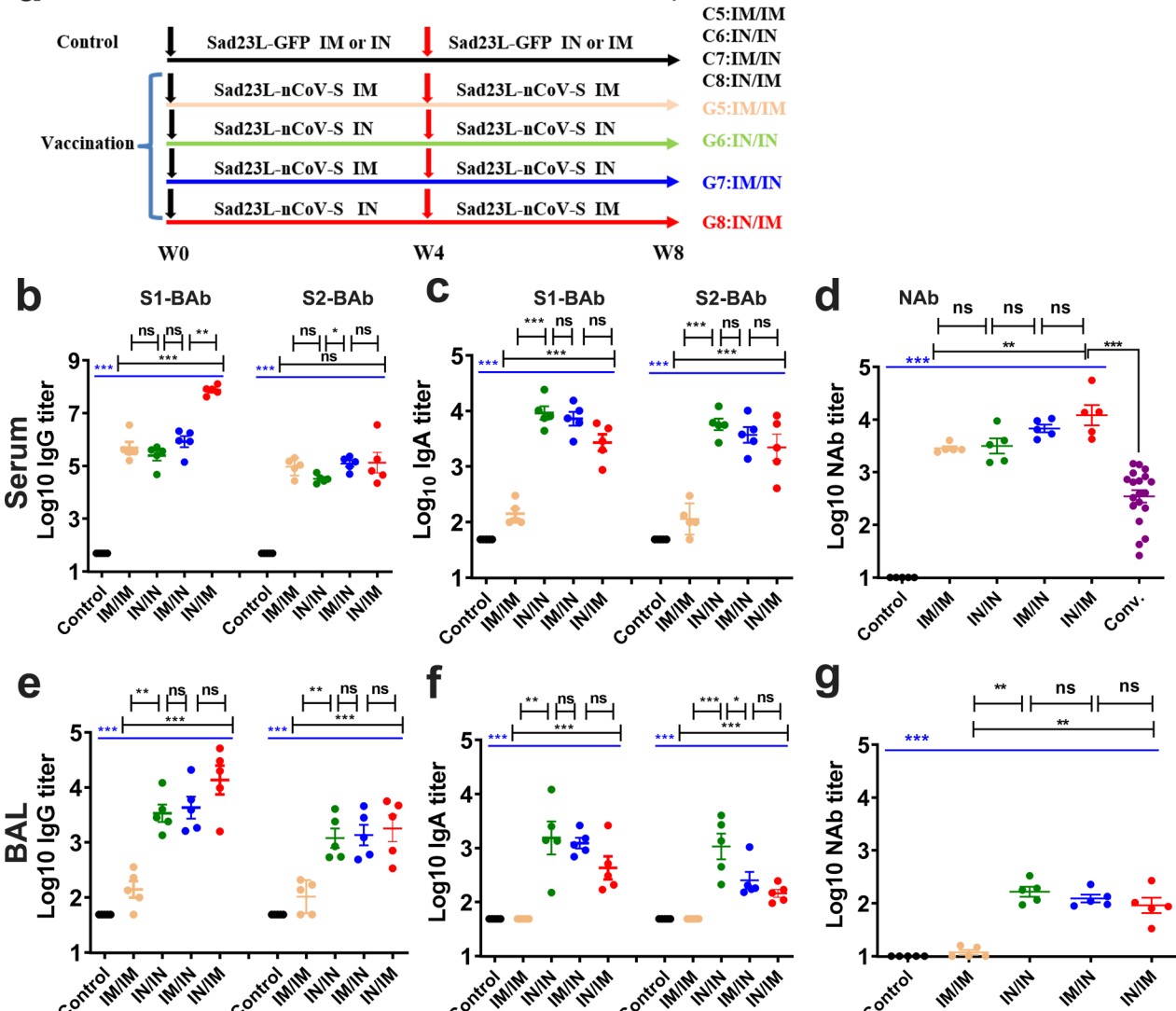

**FIG 3** Antibody responses in mice induced by four prime-boost immunization regimens with different combination of IM and IN inoculations. (a) Vaccination and control groups with prime and boost immunization regimens, including IM/IM (C5; G5), IN/IN (C6; G6), IM/IN (C7; G7), and IN/IM (C8; G8) for inoculations of Sad23L-nCoV-S vaccine or Sad23L-GFP at 4-week interval. (b and c) Serum S1-BAb or S2-BAb IgG and IgA by enzyme-linked immunosorbent assay (ELISA). (d) Serum NAb to wild-type strain of pseudovirus or convalescent serum of COVID-19 patients by pseudovirus neutralization test (pVNT). (e, f, and g) bronchoalveolar lavage (BAL) S1-BAb or S2-BAb IgG and IgA, and neutralizing antibody (NAB) to wild-type strain pseudovirus. Data were shown as mean ± SEM (standard errors of means). *P* values were analyzed by one-way ANOVA or unpaired two-tailed *t* test and Mann-Whitney test. Statistically significant differences were indicated with asterisks (*$P < 0.05$, **$P < 0.01$, and ***$P < 0.001$). ns, not significant ($P > 0.05$).

## Specific cellular immune response stimulated by prime-boost vaccination regimens with Sad23L-nCoV-S vaccine

The splenocytes and lung cells were isolated from BALB/c mice vaccinated by different combination of IN and IM inoculations of Sad23L-nCoV-S vaccine, in which specific T-cell responses were evaluated (Fig. 3a). The higher IFN-γ secreting T-cell responses to SARS-CoV-2 S-peptides were detected by ELISpot in spleen from IM/IN and IN/IM inoculations of vaccine (790 or 892 SFCs/$10^6$ cells), or in lung from IN/IN and IN/IM inoculations of vaccine (801 or 1005 SFCs/$10^6$ cells), respectively (Fig. 5a). Further to characterize the S peptides-specific CD4[+] and CD8[+] T-cell responses in lung, the resident memory T (TRM) cells expressing CD103 and CD69 markers on pulmonary CD8[+] cells were measured by flow cytometry (Fig. S3). The IN/IN vaccination regimens evidently presented the highest

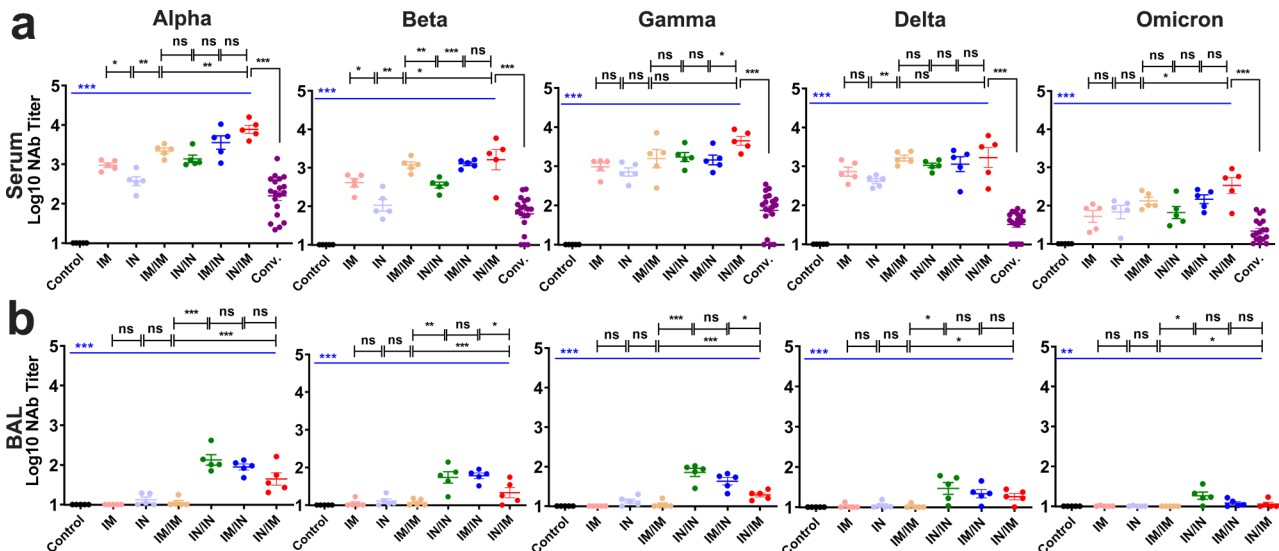

**FIG 4** Neutralization potency to five major pseudotyped SARS-CoV-2 variants by serum or bronchoalveolar lavage (BAL) NAb from different prime or prime-boost regimens vaccinated mice. Pseudotyped Alpha, Beta, Gamma, Delta, and Omicron variants of SARS-CoV-2 were used to titrate NAb activity. (a) Serum NAb and (b) BAL NAb from vaccination (IM, IN, IM/IM, IN/IN, IM/IN, and IN/IM) or control groups were presented as $IC_{50}$ titer by pseudovirus neutralization test (pVNT). Conv., indicating convalescent serum from COVID-19 patients. The mean titer was marked on each group, and *P* values were analyzed by one-way ANOVA or unpaired two-tailed *t* test and Mann-Whitney test. Statistically significant differences were indicated with asterisks (*$P < 0.05$, **$P < 0.01$, and ***$P < 0.001$). ns, not significant ($P > 0.05$).

response of TRM cells in lung that was 8.45-fold higher than the control ($P < 0.001$), while the IM/IN and IN/IM inoculations also induced robust TRM cell responses that were 4.28-fold or 4.07-fold higher than the control ($P < 0.01$) and significantly higher than the IM/IM immunization regimens ($P < 0.01$), but insignificantly varied between each other ($P > 0.05$; Fig. 5b). Specific IFN-γ, IL-2, or GrzB expressing CD4+ and CD8+ T-cell responses in spleen and lung were assessed by flow cytometry (Fig. S4), which generally presented a similar pattern with above ELISpot results from four vaccination regimens (Fig. 5a, c, and d). IN/IN immunizations induced IFN-γ, IL-2, or GrzB producing CD4+/CD8+ T-cell responses at the lowest level in spleen but the highest in lung, while IM/IM vaccination had the lowest T-cell responses in lung but the highest in spleen (Fig. 5c through e). The combination of IM/IN or IN/IM inoculations acquired high levels of T-cell responses in spleen and lung (Fig. 5; Fig. S5), indicating an optimal local and systemic cellular immunity.

## Duration of immune effectiveness in Sad23L-nCoV-S vaccinated mice

During 8 months (32 weeks) of follow-up detection, the humoral and cellular immune responses were monitored constantly from seven separate groups of mice immunized by prime or plus boost inoculations of Sad23L-nCoV-S vaccine or vector control, of which generally prime-boost vaccination regimens induced higher and longer immune response than a single prime immunization via IM or IN inoculation of vaccine (Fig. 6). Serum IgG or IgA S1-BAb, NAb and splenic IFN-γ secreting T-cell responses were maintained at higher levels (IgG titers > $10^{4.34}$, IgA >$10^{1.94}$, NAb > $10^{2.33}$, or T cells > 156 SFCs/$10^6$ cells) up to 8 months by prime-boost vaccinations (Fig. 6a through d), of which an IM-involved inoculation preferentially induced higher IgG and splenic T-cell responses (Fig. 6a and d), and an IN-involved inoculation induced higher IgA response (Fig. 6b), respectively, while the combination of IM/IN or IN/IM inoculations effectively raised higher NAb potency (Fig. 6c).

BAL antibody and lung T-cell responses from vaccinated mice were examined for mucosal and local immunity up to 8 months as well (Fig. 6e through h). BAL IgG or IgA S1-BAb, NAb and lung T cells presented a relatively high and long persisting activity in immunized mice by prime-boost vaccination (IgG titers > $10^{1.35}$, IgA >$10^{1.12}$, NAb > $10^{1.06}$,

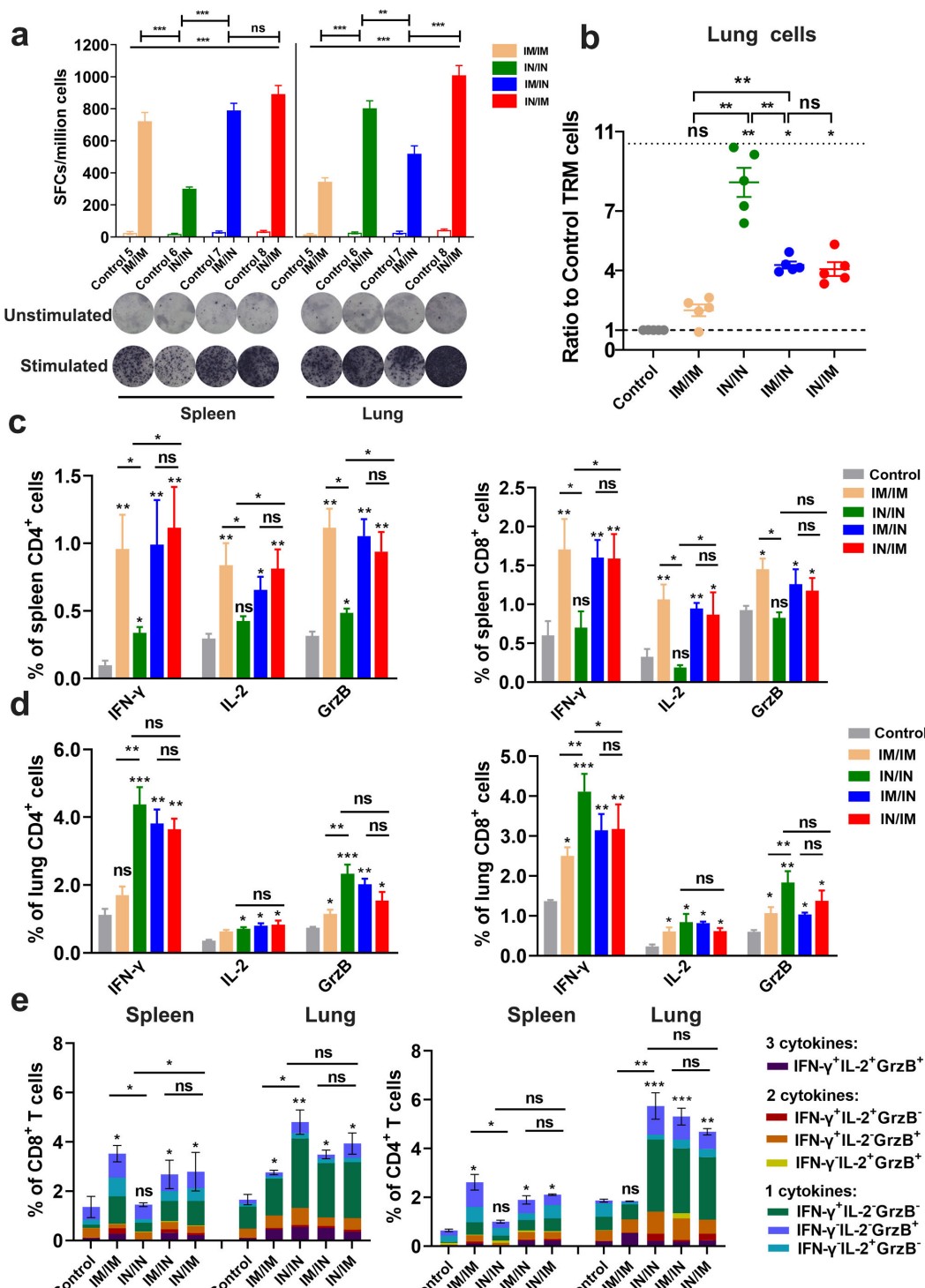

**FIG 5** Specific T-cell responses in mice induced by different prime-boost vaccination regimens with Sad23L-nCoV-S vaccine. (a) Specific IFN-γ-secreting T cells from splenic and lung cells were measured by the enzyme-linked immunospot (ELISpot) with stimulation of SARS-CoV-2 S peptides. The *Y*-axis showed the spot forming cells (SFCs) among $10^6$ cells. Representative raw data were presented for ELISpot. (b) Ratio of resident memory CD8+ T (TRM) cells (CD69+CD103+) in lung from mice induced by different prime-boost vaccination regimens or control. (c and d) Frequencies of intracellular IFN-γ, IL-2, GrzB expressing CD4+/CD8+ T-cell responses to S peptides in spleen and lung were determined by ICS, respectively. (e) The proportion of cytokine-secreting (IFN-γ+, IL-2+, or GrzB+) CD8+ or CD4+ T cells by a single, two, or three cytokines. Data shown were mean plus SEM (standard errors of means). Statistical analysis: one-way ANOVA, unpaired two-tailed *t* test and Mann-Whitney test. *$P < 0.05$, **$P < 0.01$, ***$P < 0.001$ and ns, $P > 0.05$ (not significant).

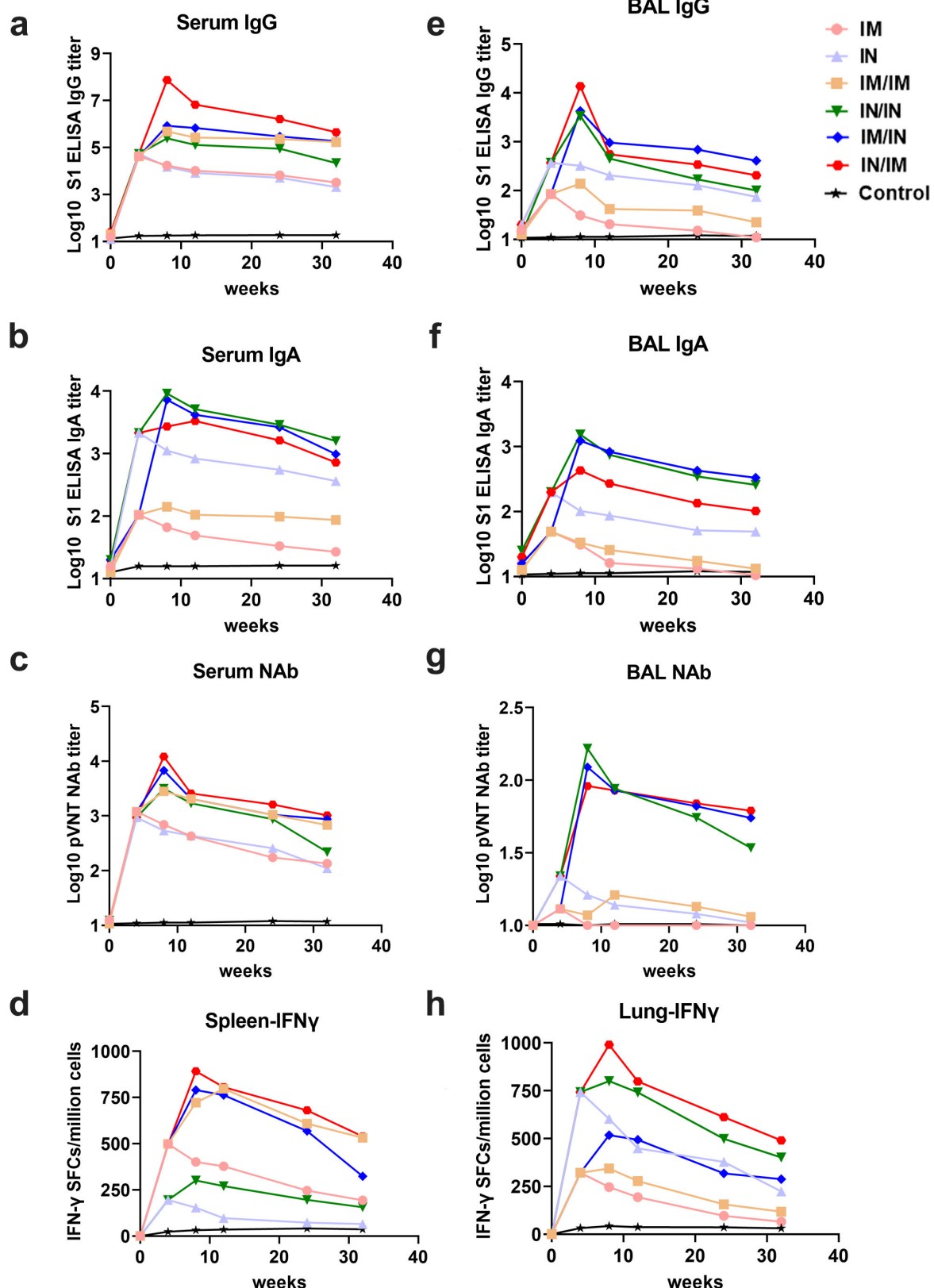

**FIG 6** Eight months follow-up detection for specific immune responses in mice immunized by different vaccination regimens with Sad23L-nCoV-S vaccine. Serum and bronchoalveolar lavage (BAL) samples were detected for antibody activity, and spleen and lung cell samples were measured for T-cell responses at five time-points of 4, 8, 12, 24, and 32 weeks post prime inoculation of vaccine. (a–d) Serum S1-BAb IgG and IgA titers, NAb $IC_{50}$ titer, and spleen IFN-$\gamma$ secreting T-cell response (SFCs/million cells). (e–h) BAL S1-BAb IgG and IgA titers, NAb $IC_{50}$ titer, and lung IFN-$\gamma$ secreting T-cell response (SFCs/million cells).

or T cells > 118 SFCs/$10^6$ cells), typically when an IN inoculation of vaccine was involved in prime-boost vaccination regimens (Fig. 6e through h).

### Neutralizing antibody to Sad23L vector in mice induced by different immunization regimens with Sad23L-nCoV-S vaccine

To evaluate the neutralizing antibody to adenovirus vector (AdNAb) in vaccinated mice, the serum and BAL samples were measured for AdNAb to Sad23L vector in 4 weeks after prime or prime-boost immunizations (Fig. S6). AdNAb titer to Sad23L increased to 1:960 in serum or 1:8 in BAL after priming with IM inoculation of Sad23L-nCoV-S vaccine (Fig. 7). In contrast, AdNAb titer was lower in serum (1:110) and similar in BAL (1:12) when primed by IN inoculation (Fig. 7), suggesting an IN inoculation could be used as primer or booster of vaccine in combination with an IM immunization by prime-boost vaccination. AdNAb level was elevated after boosting immunization of Sad23L-nCoV-S vaccine, while IN/IN and IN/IM vaccinations elicited relatively low AdNAb activity in serum or BAL of mice (Fig. 7), suggesting the combination of IN inoculation by homologous prime-boost immunization would not be affected heavily for mucosal immunity by AdNAb.

### DISCUSSION

The essential goal of vaccines is to generate potent and long-term protection against diseases. The failures to prevent SARS-CoV-2 Omicron variant infection and transmission between individuals vaccinated by most COVID-19 vaccines are considered for immune escaping of viruses (26, 27). However, the lower mucosal immunity in the upper respiratory tract of vaccinees, where Omicron targeted mainly, might be a key reason for the incomplete protection (28). Hence, mucosal IgA protection might be locally effective against Omicron infection (29). According to the draft landscape of COVID-19 candidate vaccines of World Health Organization (WHO), over 20 mucosal immune vaccines have been in clinical trials by oral, intranasal, or aerosol and inhaling inoculation (30). Two needle-free COVID-19 vaccines that are delivered through the mouth or nose have been approved for use in China and India, which are generated with adenovirus vectors by CanSino Biologics or Bharat Biotech (18). Moreover, several studies have shown the advantage of adenovirus-vectored vaccines in eliciting mucosal immunity

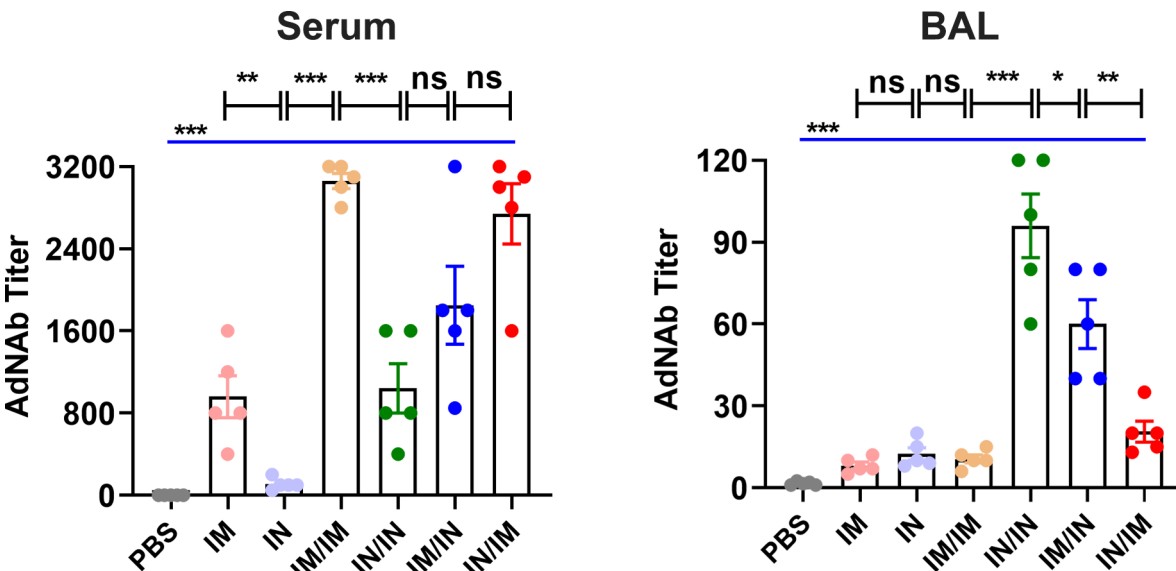

**FIG 7** Measurement of neutralizing antibody to adenovirus vector (AdNAb) in different immunization regimens with Sad23L-nCoV-S vaccine. Serum and bronchoalveolar lavage (BAL) samples were obtained in 4 weeks after prime or prime-boost immunizations, and AdNAb titer to Sad23L was measured from the vaccinated mice by different inoculation regimens, respectively. Data shown were mean plus SEM (standard errors of means). Statistical analysis: one-way ANOVA, unpaired two-tailed *t* test and Mann-Whitney test. *$P < 0.05$, **$P < 0.01$, ***$P < 0.001$ and ns, $P > 0.05$ (not significant).

with protective effects against respiratory virus infections such as Ebola virus, human respiratory syncytial virus (RSV), or MERS-CoV infection (31–33). In this study, we utilized a novel simian adenovirus 23 vectored COVID-19 vaccine (Sad23L-nCoV-S) with low pre-existing immunity in humans to evaluate the effects of systemic and local mucosal immunity in mice by different immunization regimens via IN and IM routes. The superior immunogenicity of adenovirus-vectored vaccine has been previously demonstrated by IM injection (23, 24). As shown in Fig. 2, the circulating humoral immune responses presented similar level between IM and IN prime-inoculations of Sad23L-nCoV-S vaccine, while a single IN immunization showed the specific mucosal antibody response with higher IgA and NAb in BAL and stronger T-cell response in lung of mice. The data suggested that an IN inoculation was important for eliciting mucosal immunity against virus infection at respiratory track where Omicron variant mostly invaded.

The relatively low level of immune response from a single dose of vaccine immunization suggested that a booster is needed for enhancing the humoral and cellular immunity. By prime-boost vaccinations, the factors including vaccine inoculation route, order, and combination are critical for deserving the optimal immune efficacy. It has been shown that administering of vaccine by different routes could have significantly different effects on the type and strength of the induced immune responses (34). The different inoculation routes presented the diverse immune responses of vaccine in our study. The IN/IN, IN/IM, or IM/IN inoculations of Sad23L-nCoV-S vaccine could stimulate higher protective mucosal or local immunity in BAL and lung compared with IM/IM routes (Fig. 3 to 5), and also IN/IM or IM/IN inoculations could induce higher systemic immunity in serum and spleen (Fig. 3 to 5). And long-term sustained systemic and mucosal NAb and T-cell immunity to SARS-CoV-2 were maintained at high levels over 32 weeks by prime-boost vaccination regimens in combination of IN and IM inoculations (Fig. 6). It has been reported that neutralizing antibody against adenovirus (AdNAb) can strongly reduce the immunogenicity of adenovirus-vectored vaccine (35). Our previous study indicated that heterologous prime-boost vaccinations were more effective than homologous prime-boost immunizations, because of overcoming the anti-vector immunity (23). However, the difficulties in the development of two adenovirus vectors with ideal immune effect and clinical safety limited the use of heterologous adenovirus-vectored vaccines. An evaluation showed that nasal delivery of an adenovirus-based vaccine bypassed pre-existing immunity to the vaccine vector and improved the immune response (36). Alternatively, by using the various routes for vaccinations might overcome the pre-existing immunity of adenovirus-vectored vaccine (37, 38). In this investigation, IN/IM or IM/IN elicited higher antibody titer than IM/IM did (Fig. 3), suggesting that IN boost vaccination with Sad23L-nCoV-S vaccine could bypass antivector immune responses induced by the IM prime vaccination, leading to significant increases in local mucosal and systemic immune responses. Based on the immune responses with no significant difference between IN/IM and IM/IN vaccination regimens, IN priming and IM boosting inoculations were also recommended, which could rapidly increase the local mucosal antibody response to prevent virus infection after priming immunization.

There were two limitations in this study. Firstly, the lack of live SARS-CoV-2 for measuring neutralizing antibody and challenging of vaccinated animals for vaccine efficacy by the IN involved vaccination regimens due to biosafety level-3 requirement. Secondly, the difficulty to isolate a large amount of immune cells from lung tissue of vaccinated mice for comprehensively investigating local cellular immunity. Our future studies will mechanistically investigate the kinetics of intranasal immunization through isolation of tissues in closer proximity to the nasal cavity, including mucosa-associated lymph node tissue (MALT).

In conclusion, this study illustrated the superior potential for prime-boost vaccination regimens in combination of IN and IM (IN/IM or IM/IN) inoculations of Sad23L-nCoV-S vaccine for inducing strong mucosal and systemic immunity in mice, which provided an important reference for respiratory virus vaccination regimens.

## MATERIALS AND METHODS

### Adenovirus-vectored vaccines and mice

Sad23L, a replication defective novel adenovirus vector was constructed and produced as described previously (39), and Ad5 vector (pBHGloxdeltaE13Cre) plasmid was provided by Dr. JH Zhou (State Key Laboratory of Veterinary Biotechnology, Harbin Veterinary Research Institute of Chinese Academy of Agricultural Sciences), which was initially generated by Ng et al. (40). The full-length S protein gene of SARS-CoV-2 (GenBank: MN908947.3) was optimized and synthesized (Beijing Genomics Institute, China), then was cloned into Sad23L and Ad5 vectors, respectively. The recombinant adenoviral constructs Sad23L-nCoV-S and Ad5-nCoV-S were rescued from HEK293A packaging cells (23). The Sad23L-nCoV-S and Ad5-nCoV-S vaccine strains were stained with 2% phosphotungstic acid and morphologically observed by TEM (JEM-1400, JEOL, Japan).

BALB/c mice were obtained from the Animal Experimental Centre of Southern Medical University, Guangdong, China. All animal experiments were conducted in compliance with the guidelines for the care and use of laboratory animals and approved by the Southern Medical University (SMU) Animal Care and Use Committee, Guangzhou, China (permit numbers: SYXK [Yue] 2021–0167).

### Animal immunization

Female BALB/c mice ($n$ = 18/group for 5–6 weeks) were individually inoculated with a single dose of $10^9$ PFU/100 µlL Sad23L-nCoV-S or Ad5-nCoV-S vaccine by IN drip or IM injection inoculation, respectively. A dose of $10^9$ PFU/100 µlL Sad23L-GFP or Ad5-GFP vectorial viruses was used as sham control.

Female BALB/c mice ($n$ = 15/group for 5–6 weeks) were primed with a dose of $10^9$ PFU Sad23L-nCoV-S vaccine, and then at 4-week interval boosted with a same dose of vaccine by IM/IM, IN/IN, IM/IN, or IN/IM immunization regimens, respectively. A dose of $10^9$ PFU Sad23L-GFP vectorial viruses was used as sham control.

A group of five mice were sacrificed at week 4 from a single dose of vaccine immunization and at week 8 from prime-boost immunizations for detection of systemic and mucosal immune responses. A group of three mice were sacrificed at 5 time-points of 4, 8, 12, 24, and 32 weeks post prime inoculation of vaccine for follow-up detection.

### COVID-19 patients' serum samples

A total of 21 convalescent serum samples of COVID-19 patients were kindly provided by Shenzhen Center for Disease Control and Prevention (CDC), China (41). All serum samples were heat-inactivated at 56°C for 30 min prior to use. All individual samples were aliquoted to several vials and stored at −80°C. This study was approved by the Medical Ethics Committee of Southern Medical University and Shenzhen CDC and followed the ethical guidelines of the 1975 Declaration of Helsinki.

### Preparation of serum and bronchoalveolar lavage fluid samples

The eyeballs of mice were removed, and blood was collected from the eye socket. Then, mice were sacrificed and sterilized by alcohol, and the thoracic cavity was opened to expose the lung. Bronchoalveolar lavage fluid (BAL) of mice was collected by washes with PBS by tracheal intubation and centrifuged at 1,500 rpm for 5 min. The serum and BAL samples were stored at −80°C for testing.

### Detection of SARS-CoV-2 S protein antigen and binding antibody

HEK293 cells infected with Sad23L-nCoV-S and Ad5-nCoV-S vaccine strains or vector controls were fixed in 4% cold paraformaldehyde, blocked, and incubated with mouse monoclonal antibody to SARS-CoV-2 S protein (Sino Biological, China) in 1% BSA/PBST at

4°C overnight. Anti-mouse IgG-Dylight 594 antibody (Thermo Scientific, USA) was added to the cells for 30 min at 37°C. DAPI was used for nuclei staining. The S protein expression from cell lysates was analyzed by Western blot using an anti-SARS-CoV-2 RBD polyclonal antibody (Sino Biological, China) and GAPDH as a loading control.

ELISA was used to detect the binding antibodies to S protein of SARS-CoV-2 (S-BAb). The microtiter plates (Corning, USA) were coated overnight at 4°C with 1 µg/ml of SARS-CoV-2 S1 (Sino Biological, 40591-V08H) or S2 proteins (Sino Biological, 40590-V08H1). Serum and BAL samples were threefold serially diluted, and anti-S1 and S2 binding antibodies (S1-BAb and S2-BAb; S-BAb) were tested by ELISA. Secondary antibodies of goat anti-mouse IgA-HRP (Abcam, UK) or rabbit anti-mouse IgG-HRP (Bioss, China) at 1:4000 dilution were incubated at 37°C. The absorbance at 450 nm was recorded using a BioTek Epoch. The endpoint serum or BAL dilution was calculated by a curve fit analysis of optical density (OD) values for serially diluted sera with a signal than cut-off value (S/CO) ≥3.

## Pseudovirus neutralization test

Pseudoviruses expressing SARS-CoV-2 S protein of Wild-type (Wuhan-1) strain, Alpha (B.1.1.7), Beta (B.1.351), Gamma (P.1), Delta (B.1.617.2), or Omicron (B.1.1.529) variants were generated, and the neutralizing antibody (NAb) to SARS-CoV-2 was measured by pseudovirus neutralization test (pVNT) (24). Briefly, the psPAX2 (Addgene), pLenti-CMV Puro-Luc (Addgene), and pcDNA3.1-SARS-CoV-2 S plasmids were co-transfected into HEK-293T cells. After 48 h, the supernatants containing pseudovirus were collected, filtered through 0.45 µm filter, aliquoted, and stored at −80°C. The neutralizing activity of the mouse serum and BAL samples was determined. Twofold serial dilutions of heat inactivated serum and BAL were prepared and mixed with 50 µl pseudovirus. The mixture was incubated at 37°C for 1 h and added to $3 \times 10^4$ HEK293T-hACE2 cells (Sino Biological). After 48 h incubation, the cells were lysed by Bright-Glo luciferase assay (Promega) according to the manufacturer's instructions. NAb titer ($IC_{50}$) to SARS-CoV-2 was defined as the sample dilution at which a 50% reduction (50% inhibitory concentration) in relative luciferase activity (RLU) observed relative to the average of the virus control wells.

## Enzyme-linked immunospot

Specific IFN-γ secreting T-cell response was measured by the enzyme-linked immunospot (ELISpot). Mouse spleen was grinded and filtered through a 70 µm cell strainer, which was centrifuged and resuspended in 5 mL ACK lysing buffer for 5 min at room temperature, diluted in 10 mL PBS and centrifuged. The single-cell suspensions were resuspended in 1 mL complete RPMI-1640 medium. The lungs were cut into small pieces and digested with collagenase IV (2 mg/mL) and DNase I (5 mg/mL) at 37°C for 60 min, and then filtered with 70 µm cell strainers. After the red blood cells were lysed, PBS was added and washed twice, and finally resuspended with 1 mL complete RPMI-1640 medium.

Mouse splenocytes and lung cells ($5 \times 10^5$ cells/well) were stimulated with S peptides (5 µg/ml) in duplicate. SARS-CoV-2 S antigen-specific T lymphocyte response in mice was assessed by IFN-γ ELISpotPLUS kits (Mabtech) following the manufacturer's instruction. S peptides were predicted (http://www.iedb.org/) and synthesized by Guangzhou IGE Biotechnology LTD. Spots were counted with a CTL Immunospot Reader (Cellular Technology Ltd). The results were expressed as spot forming cells (SFCs) per million cells.

## Cytokine staining and flow cytometry

The mouse splenocytes and lung cells ($8 \times 10^5$ cells/well) were stimulated with S peptides (5 µg/ml) or medium as negative control for 12 h at 37°C before a 6 h treatment with brefeldin A (BD Pharmingen, 554724). Cells were collected and stained with antibodies to

mouse CD45, CD4, and CD8 surface markers (BD). The cells were washed twice with PBS and fixed with IC fixation buffer, permeabilized with permeabilization buffer (BD) and stained with antibodies to mouse interferon-γ (IFN-γ), interleukin-2 (IL-2), tumor necrosis factor α (TNF-α), and GrzB (BD). A part of the lung cells was stained with antibodies to mouse CD45, CD4, CD8, CD103, and CD69 surface markers after stimulation. Boolean gating was applied to the gated IFN-γ$^+$, IL-2$^+$, GrzB$^+$ cells to determine the frequency of all cytokine$^+$ CD4$^+$ and CD8$^+$ T cells. All samples were tested with a BD FACSCanto Flow Cytometer (BD) and the data were analyzed using FlowJo Software (version 10).

## Adenovirus neutralizing antibody assay

Mouse serum and BAL samples were collected and tested on HEK-293A cells for neutralizing Sad23L-GFP viruses by green fluorescent activity assay. Adenovirus neutralization antibody (AdNAb) titers were defined as the maximum serum or BAL dilution that neutralized 50% of Green activity as previously described (42).

## Statistical analyses

Statistical significance between groups was analyzed using one-way analysis of variance or unpaired two-tailed $t$ test and Mann-Whitney test. The correlation between S-BAb and NAb titers was analyzed using Pearson's correlation coefficients. Statistically significant differences were indicated with asterisks (*$P < 0.05$, **$P < 0.01$, and ***$P < 0.001$). All graphs were generated with GraphPad Prism 8 software.

## ACKNOWLEDGMENTS

The authors thank Shenzhen CDC for providing COVID-19 patients' serum samples.

This work was supported by grants from the National Natural Science Foundation of China (No. 32070929, 82271868, and 31970886), Guangdong Natural Science Foundation Outstanding Youth Project (No. 2022B1515020050), Provincial Key Laboratory of Immune Regulation and Immunotherapy (2022B1212010009) and Guangzhou Bai Rui Kang (BRK) Biological Science and Technology Limited Company (Guangzhou, China), Guangdong Basic and Applied Basic Research Foundation (2021A1515110991) and Research Foundation of Shenzhen Hospital of Southern Medical University (PT2021GZR09).

## AUTHOR AFFILIATIONS

[1]Department of Transfusion Medicine, School of Laboratory Medicine and Biotechnology, Southern Medical University, Guangzhou, China
[2]Guangzhou Bai Rui Kang (BRK) Biological Science and Technology Limited Company, Guangzhou , China
[3]Department of Pediatrics, Shenzhen Hospital, Southern Medical University, Shenzhen, China
[4]Shenzhen Bao'an District Central Blood Station, Shenzhen, China
[5]Department of Bioengineering, School of Medicine and College of Engineering, University of Washington, Seattle, Washington, USA

## AUTHOR ORCIDs

Chengyao Li ⬩ http://orcid.org/0000-0002-2087-873X
Tingting Li ⬩ http://orcid.org/0000-0001-5727-2179

## FUNDING

| Funder | Grant(s) | Author(s) |
| --- | --- | --- |
| MOST \| National Natural Science Foundation of China (NSFC) | 31970886 | Tingting Li |

| Funder | Grant(s) | Author(s) |
|---|---|---|
| GDSTC \| Natural Science Foundation of Guangdong Province (廣東省自然科學基金) | 2022B1515020050 | Tingting Li |
| MOST \| National Natural Science Foundation of China (NSFC) | 32070929 | Chengyao Li |
| MOST \| National Natural Science Foundation of China (NSFC) | 82271868 | Chengyao Li |
| GDSTC \| Basic and Applied Basic Research Foundation of Guangdong Province (廣東省基礎與應用基礎研究專項資金) | 2021A1515110991 | Shengxue Luo |

## AUTHOR CONTRIBUTIONS

Panli Zhang, Data curation, Investigation, Validation, Visualization, Writing – original draft | Shengxue Luo, Formal analysis, Funding acquisition, Investigation | Peng Zou, Formal analysis, Visualization | Qitao Deng, Investigation | Cong Wang, Formal analysis | Jinfeng Li, Resources | Peiqiao Cai, Validation | Ling Zhang, Project administration | Chengyao Li, Funding acquisition, Supervision, Writing – review and editing | Tingting Li, Funding acquisition, Supervision, Writing – review and editing

## DATA AVAILABILITY

All data are available from the main article or supplemental information.

## ADDITIONAL FILES

The following material is available online.

## Supplemental Material

**Fig. S1-S6 (Spectrum01794-23_S0001.pdf).** Supplemental material.

## Open Peer Review

**PEER REVIEW HISTORY (review-history.pdf).** An accounting of the reviewer comments and feedback.

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
