## [Reviewer comments · Microbiology Spectrum]

Microbiology Spectrum

A novel simian adenovirus vectored COVID-19 vaccine elicits effective mucosal and systemic immunity in mice by intranasal and intramuscular vaccination regimens

Panli Zhang, Shengxue Luo, Peng Zou, Qitao Deng, Cong Wang, Jinfeng Li, Peiqiao Cai, Ling Zhang, Chengyao Li, and Tingting Li

Corresponding Author(s): Tingting Li, Southern Medical University

Review Timeline:

Submission Date:	April 28, 2023
Editorial Decision:	August 12, 2023
Revision Received:	September 11, 2023
Accepted:	September 19, 2023

Editor: Bo Zhang

Reviewer(s): The reviewers have opted to remain anonymous.

Transaction Report:

DOI: <https://doi.org/10.1128/spectrum.01794-23>

August 12, 2023

Dr. Tingting Li
Southern Medical University
Transfusion Medicine
No,1838 North Guangzhou Avenue
Guangzhou, Guangdong 510515
China

Re: Spectrum01794-23 (A novel simian adenovirus vectored COVID-19 vaccine elicits effective mucosal and systemic immunity in mice by intranasal and intramuscular vaccination regimens)

Dear Dr. Tingting Li:

Link Not Available

Sincerely,

Bo Zhang

Journals Department
Reviewer comments:

Reviewer #1 (Comments for the Author):

Major concern:

1. The introduction section states that the chimpanzee adenovirus Choxad1 has poor respiratory immunity in clinical trials. Does Sad23 also have this issue? What is the infectivity of this virus to respiratory cells, and has the receptor been verified?
2. Although it has been mentioned in the conclusion that developing heterologous adenovirus vectors at the same time poses

certain difficulties, have the authors considered the effectiveness using Sad23 for secondary immunization after the initial immunization of the current Ad5-cov vaccine.

3. The article mentioned that, using the IN immunization method, lower levels of Sad Neutralizing antibody can be produced in the mouse serum. Have the authors conducted any experiments to verify whether Sadv could infect mouse cells?

4. What are the advantages of Sadv over Ad5?-

Minor concerns:

1. In Figure 1c, compared with the control group, did cells in Sad23L-nCoV-S and Ad5-nCoV-S show significant CPE after infection?
2. In Figure 1d, Sad23L-nCoV-S detected the expression of S1 and S proteins, but the expression of S protein in Ad5-nCoV-S was not obvious.
3. In Figure 4, it is suggested to show the neutralization result of pseudovirus after a single immunization (IN/IM), further indicating that booster needle can increase neutralizing antibodies;
4. It is recommended to perform the neutralization assay and challenge test using live SARS-CoV-2.

Reviewer #2 (Comments for the Author):

Adenovirus vectors are the most commonly used viral vector for COVID-19 vaccines. One of the major drawbacks of adenovectors is the pre-existing immunity against the vectors, which can limit the effectiveness of vaccination. In this study, Zhang et al constructed and evaluated a novel simian adenovirus vectored COVID-19 vaccine. They showed that their new adenovector, Sad23L could elicit potent and durable humoral and cellular immuno-response through either IN/IM or IM/IN prime-boost vaccination regimens.

Major points:

1. As shown in Fig. 1d, The S protein abundance in Sad23L-infected cells is apparently more than that in Ad5-infected cells. This could have an unneglectable impact on the immunogenicity of the two vaccines.
2. The major advantage of Sad23L is to bypass Ad5 pre-existing immunity. However, the authors did not compare the immunogenicity of Sad23L-vaccine between normal mice and Ad5-pre-exposed mice in this study.

Minor points:

1. P63, breakthrough infections are commonly seen...
2. P60, The authors are suggested to describe why mRNA vaccine can only be administrated through IM route and thus fails to elicit comparable mucosal immunity to the nasal vaccines.
3. In Fig. 3B, IN/IM elicited higher S1-BAb IgG titer in the serum than IM/IM did. The authors need to address this phenotype in the discussion section.
4. No statistical analysis for Fig. 7. Otherwise, it is impossible to conclude that IN boost could bypass the anti-vector immunity (Line 310-312).
5. Line 295-297, the authors stated that IN/IM or IM/IN can induce more potent systemic immunity than IM/IM. However, Fig. 3D shows IM/IN cannot induce more Nab than IM/IM in the serum.
6. English writing needs to be professionally polished.

Staff Comments:

Preparing Revision Guidelines

- Point-by-point responses to the issues raised by the reviewers in a file named "Response to Reviewers," NOT IN YOUR COVER LETTER.
- Upload a compare copy of the manuscript (without figures) as a "Marked-Up Manuscript" file.
- Each figure must be uploaded as a separate file, and any multipanel figures must be assembled into one file.
- Manuscript: A .DOC version of the revised manuscript

- Figures: Editable, high-resolution, individual figure files are required at revision, TIFF or EPS files are preferred

Please return the manuscript within 60 days; if you cannot complete the modification within this time period, please contact me. If you do not wish to modify the manuscript and prefer to submit it to another journal, please notify me of your decision immediately so that the manuscript may be formally withdrawn from consideration by Microbiology Spectrum.

Adenovirus vectors are the most commonly used viral vector for COVID-19 vaccines. One of the major drawbacks of adenovectors is the pre-existing immunity against the vectors, which can limit the effectiveness of vaccination. In this study, Zhang et al constructed and evaluated a novel simian adenovirus vectored COVID-19 vaccine. They showed that their new adenovector, Sad23L could elicit potent and durable humoral and cellular immuno-response through either IN/IM or IM/IN prime-boost vaccination regimens.

Major points:

1. As shown in Fig. 1d, The S protein abundance in Sad23L-infected cells is apparently more than that in Ad5-infected cells. This could have an unneglectable impact on the immunogenicity of the two vaccines.
2. The major advantage of Sad23L is to bypass Ad5 pre-existing immunity. However, the authors did not compare the immunogenicity of Sad23L-vaccine between normal mice and Ad5-pre-exposed mice in this study.

Minor points:

1. P63, breakthrough infections are commonly seen...
2. P60, The authors are suggested to describe why mRNA vaccine can only be administrated through IM route and thus fails to elicit comparable mucosal immunity to the nasal vaccines.
3. In Fig. 3B, IN/IM elicited higher S1-BAb IgG titer in the serum than IM/IM did. The authors need to address this phenotype in the discussion section.
4. No statistical analysis for Fig. 7. Otherwise, it is impossible to conclude that IN boost could bypass the anti-vector immunity (Line 310-312).
5. Line 295-297, the authors stated that IN/IM or IM/IN can induce more potent systemic immunity than IM/IM. However, Fig. 3D shows IM/IN cannot induce more Nab than IM/IM in the serum.
6. English writing needs to be professionally polished.

Reviewer #1:

Major concern:

1. The introduction section states that the chimpanzee adenovirus ChAdox1 has poor respiratory immunity in clinical trials. Does Sad23 also have this issue? What is the infectivity of this virus to respiratory cells, and has the receptor been verified?

Answer: Thanks for your questions. Sad23L and ChAdox1 have similarly poor respiratory immune responses through a single dose of intramuscular or intranasal inoculation of vaccines. In this manuscript, we proposed a prime-boost vaccination regimens to induce mucosal and systematic immunity by combining intramuscular and intranasal inoculations of vaccines. The infectivity of Sad23L and ChAdox1 has not been well identified. Recently, we examined the infectivity of Sad23L-luc in respiratory cells of mice after intranasal inoculation of virus by bioluminescence imaging, which showed that Sad23L vector strain could infect the nasal mucosal cells where the Sad23L-luc drip inoculated (R-Figure 1 for reviewing only).

R-Figure 1 for reviewing only. Detection of Sad23L-luc in nasal mucosal cells of mouse nose by bioluminescence imaging.

2. Although it has been mentioned in the conclusion that developing heterologous adenovirus vectors at the same time poses certain difficulties, have the authors considered the effectiveness using Sad23 for secondary immunization after the initial immunization of the current Ad5-cov vaccine.

Answer: Thank you for your comment and suggestion. In our previous study, we found that Sad23L had no cross-reactivity with Ad5 and could elicit strong immunity in the Ad5 pre-exposed mice (Luo S, et al. *Virus Res.* 2019;268:1-10). According to previous publications regarding adenoviruses (Ad5/Ad26) vectored vaccines (Logonov DY, et al. *Lancet.* 2020;396(10255):887-897), the novel adenovirus vector Ad26 could be for secondary immunization after exposure of Ad5, which produced robust and long-lasting immune response. Therefore, we supposed that Sad23 vectored vaccine could be used for secondary immunization after the initial immunization of the current Ad5-cov vaccine.

3. The article mentioned that, using the IN immunization method, lower levels of Sad

Neutralizing antibody can be produced in the mouse serum. Have the authors conducted any experiments to verify whether Sadv could infect mouse cells?

Answer: Thanks for your question. In our previous studies, we constructed the novel adenoviral vector carrying a firefly luciferase reporter gene (Sad23L-luc). After intravenous injection of this vector into mice, Sad23-luc was found in multiple organs (R-Figure 2) by bioluminescence imaging, while Sad23 DNA was detected in lung, spleen and liver tissues (R-Figure 3) by PCR.

R-Figure 2 for reviewing only. Biodistribution of the Sad23-luc vector in mice after intravenous injection. Sad23-luc inoculated mice at both sides and blank control at the middle.

R-Figure 3 for reviewing only. PCR detection of Sad23L DNA in various tissues of Sad23L-nCoV-S immunized mice.

4. What are the advantages of Sadv over Ad5?-

Answer: Thanks for your question. In our previous study, we found the pre-existing antibody of Sad23 was significantly lower than that of Ad5 among 600 healthy Chinese blood donors. The seroprevalence of Sad23L and Ad5 was 10.2% and 75.2%, respectively (Luo S, et al. *Emerg Microbes Infect.* 2021;10(1):1002-1015). The major advantage of Sad23L is to bypass Ad5 pre-existing immunity.

Minor concerns:

1. In Figure 1c, compared with the control group, did cells in Sad23L-nCoV-S and Ad5-nCoV-S show significant CPE after infection?

Answer: There was significant CPE in Sad23L-nCoV-S and Ad5-nCoV-S infected cells but not in the PBS control cells.

2. In Figure 1d, Sad23L-nCoV-S detected the expression of S1 and S proteins, but the expression of S protein in Ad5-nCoV-S was not obvious.

Answer: Thank you very much for your comment. In the previous Figure 1d, the S band was not fully selected from Ad5-nCoV-S lanes. In the revised manuscript, Fig.1 d has been replaced by the newly plotted Figure 1d with full selection of S bands and controls.

3. In Figure 4, it is suggested to show the neutralization result of pseudovirus after a single immunization (IN/IM), further indicating that booster needle can increase neutralizing antibodies;

Answer: Thank you very much for your comment. Indeed, Figure 4 shows that a booster of IM or IN inoculation can increase neutralizing antibodies after a single IN or IM immunization.

4. It is recommended to perform the neutralization assay and challenge test using live SARS-CoV-2.

Answer: Thank you for your suggestion. For the sake of biosafety control, the live SARS-CoV-2 is almost banned in China for the classical neutralization assay and challenge test in the experimental study. Alternatively, the pseudotyped SARS-CoV-2 (pseudovirus) neutralization test (pVNT) is commonly used to measure the NAb reactivity in the BSL-2 laboratory. The large numbers of publications have demonstrated a close correlation between pseudovirus and live virus neutralization assays for measuring NAb from adenovirus vectored COVID-19 vaccines immunized rhesus macaques ($P < 0.0001$, $R = 0.8427$, Mercado NB, et al. *Nature*. 2020;586(7830):583-588; $P < 0.0001$, $R = 0.8052$, Yu J, et al. *Science*. 2020;369(6505):806-811) and humans ($P < 0.0001$, $R = 0.72$, Zhu F, et al. *Lancet*. 2020;396(10249):479-488).

Reviewer #2:

Adenovirus vectors are the most commonly used viral vector for COVID-19 vaccines. One of the major drawbacks of adenovectors is the pre-existing immunity against the vectors, which can limit the effectiveness of vaccination. In this study, Zhang et al constructed and evaluated a novel simian adenovirus vectored COVID-19 vaccine. They showed that their new adenovector, Sad23L could elicit potent and durable humoral and cellular immuno-response through either IN/IM or IM/IN prime-boost vaccination regimens.

Major points:

1. As shown in Fig. 1d, The S protein abundance in Sad23L-infected cells is apparently more than that in Ad5-infected cells. This could have an unneglectable impact on the immunogenicity of the two vaccines.

Answer: Thank very much for your careful observation and comment. In previous Fig. 1d with the representative Western blot image, it looks the S protein band from Sad23L vaccine is darker than that from Ad5 vaccine. In the study, we used an equal dose of Sad23L or Ad5 vectored vaccine for vaccination, and the results were comparable. In the revised manuscript, Fig. 1d has been replaced by the newly plotted

Western blot with full selection of S bands and controls.

2. The major advantage of Sad23L is to bypass Ad5 pre-existing immunity. However, the authors did not compare the immunogenicity of Sad23L-vaccine between normal mice and Ad5-pre-exposed mice in this study.

Answer: Thank you for your comment. In our previous study, we evaluated Sad23L-prM-E and Ad5-prM-E ZIKV vaccines in mice with or without Ad5 pre-exposure. As shown in the following figure, ZIKV E-specific antibody T cell responses were similar from Sad23L-vaccine but significantly not different from Ad5-vaccine between normal mice and Ad5-pre-exposed mice (Luo S, et al. Virus Res. 2019;268:1-10). The results suggested that Ad5 pre-exposure did not affect the immunogenicity of Sad23L vectored vaccine.

Minor points:

1. P63, breakthrough infections are commonly seen...

Answer: The sentence has been revised (page 4 and line 62).

2. P60, the authors are suggested to describe why mRNA vaccine can only be administrated through IM route and thus fails to elicit comparable mucosal immunity to the nasal vaccines.

Answer: Thanks for your suggestion. The description is added in the introduction section in the revised manuscript (Page 4 and line 60).

3. In Fig. 3B, IN/IM elicited higher S1-BAb IgG titer in the serum than IM/IM did. The authors need to address this phenotype in the discussion section.

Answer: Thanks for your suggestion. The description has been added in the discussion section of revised manuscript (Page 15 and line 305).

4. No statistical analysis for Fig. 7. Otherwise, it is impossible to conclude that IN boost could bypass the anti-vector immunity (Line 310-312).

Answer: Statistical analysis was added in Figure 7.

5. Line 295-297, the authors stated that IN/IM or IM/IN can induce more potent systemic immunity than IM/IM. However, Fig. 3D shows IM/IN cannot induce more Nab than IM/IM in the serum.

Answer: Thanks for your comment. In fact, the NAbs induced by IM/IN are

slightly higher than IM/IM in the serum ($10^{3.83}$ vs $10^{3.46}$, Fig. 3d). In addition, the IFN- γ secreting T cell responses to SARS-CoV-2 S-peptides in spleen were detected higher from IM/IN than those from IM/IM (790 vs 722 SFCs/ 10^6 cells, Fig. 5a). Therefore, we stated that IN/IM or IM/IN can induce more potent systemic immunity than IM/IM.

6. English writing needs to be professionally polished.

Answer: Thank you for your suggestion. The new version of manuscript has been carefully and thoroughly edited for English writing.

September 19, 2023

Dr. Tingting Li
Southern Medical University
Transfusion Medicine
No,1838 North Guangzhou Avenue
Guangzhou, Guangdong 510515
China

Re: Spectrum01794-23R1 (A novel simian adenovirus vectored COVID-19 vaccine elicits effective mucosal and systemic immunity in mice by intranasal and intramuscular vaccination regimens)

Dear Dr. Tingting Li:

Your manuscript has been accepted, and I am forwarding it to the ASM Journals Department for publication. You will be notified when your proofs are ready to be viewed.

Sincerely,

Bo Zhang
Editor, Microbiology Spectrum
